# Postmortem Cardiopulmonary Pathology in Patients with COVID-19 Infection: Single-Center Report of 12 Autopsies from Lausanne, Switzerland

**DOI:** 10.3390/diagnostics11081357

**Published:** 2021-07-28

**Authors:** Sabina Berezowska, Karine Lefort, Kalliopi Ioannidou, Daba-Rokhya Ndiaye, Damien Maison, Constantinos Petrovas, Samuel Rotman, Nathalie Piazzon, Dina Milowich, Nathalie Sala, Chun-Yi Tsai, Eleonore Multone, Pierre-Yves Bochud, Mauro Oddo, Bettina Bisig, Laurence de Leval

**Affiliations:** 1Department of Laboratory Medicine and Pathology, Institute of Pathology, Lausanne University Hospital, University of Lausanne, Rue du Bugnon 25, 1011 Lausanne, Switzerland; karine.lefort@chuv.ch (K.L.); kalliopi.ioannidou@chuv.ch (K.I.); rokhya.ndiaye91@gmail.com (D.-R.N.); damien.maison@chuv.ch (D.M.); Konstantinos.petrovas@chuv.ch (C.P.); samuel.rotman@chuv.ch (S.R.); nathalie.piazzon@chuv.ch (N.P.); dina.milowich@hopitalvs.ch (D.M.); nathalie.sala@chuv.ch (N.S.); cyt3021@gmail.com (C.-Y.T.); eleonore.multone@gmail.com (E.M.); bettina.bisig@chuv.ch (B.B.); laurence.deleval@chuv.ch (L.d.L.); 2Department of Medicine, Service of Infectious Diseases, Lausanne University Hospital, University of Lausanne, Rue du Bugnon 46, 1011 Lausanne, Switzerland; Pierre-Yves.Bochud@chuv.ch; 3Adult Intensive Care Medicine Service, Lausanne University Hospital, University of Lausanne, Rue du Bugnon 46, 1011 Lausanne, Switzerland; mauro.oddo@chuv.ch; 4Medical Direction for Research and Education, Lausanne University Hospital, University of Lausanne, Rue du Bugnon 21, 1011 Lausanne, Switzerland

**Keywords:** SARS-CoV-2, COVID-19, diffuse alveolar damage, immunohistochemistry, RNAscope, reverse transcriptase quantitative PCR, autopsy, post-mortem diagnostics

## Abstract

We report postmortem cardio-pulmonary findings including detection of severe acute respiratory syndrome coronavirus 2 (SARS-CoV-2) in formalin-fixed paraffin embedded tissue in 12 patients with COVID-19. The 5 women and 7 men (median age: 73 years; range 35–96) died 6–38 days after onset of symptoms (median: 14.5 days). Eight patients received mechanical ventilation. Ten patients showed diffuse alveolar damage (DAD), 7 as exudative and 3 as proliferative/organizing DAD. One case presented as acute fibrinous and organizing pneumonia. Seven patients (58%) had acute bronchopneumonia, 1/7 without associated DAD and 1/7 with aspergillosis and necrotic bronchitis. Microthrombi were present in 5 patients, only in exudative DAD. Reverse transcriptase quantitative PCR detected high virus amounts in 6 patients (50%) with exudative DAD and symptom-duration ≤14 days, supported by immunohistochemistry and in-situ RNA hybridization (RNAscope). The 6 patients with low viral copy levels were symptomatic for ≥15 days, comprising all cases with organizing DAD, the patient without DAD and one exudative DAD. We show the high prevalence of DAD as a reaction pattern in COVID-19, the high number of overlying acute bronchopneumonia, and high-level pulmonary virus detection limited to patients who died ≤2 weeks after onset of symptoms, correlating with exudative phase of DAD.

## 1. Introduction

During the last one and a half years, Coronavirus disease 2019 (COVID-19) has emerged as a worldwide pandemic, caused by the novel severe acute respiratory syndrome coronavirus 2 (SARS-CoV-2). Although many patients present with only mild symptoms such as cough, chills and fatigue, and some are asymptomatic, around 20% develop severe shortness of breath leading to respiratory failure [1]. The disease mainly affects the lungs, most often leading to death due to acute respiratory distress syndrome (ARDS). 

Taken into account the high number of deaths resulting from the disease worldwide (>3,404,000 until 19 May 2021) [2], documentation of the histological lesions is still relatively limited, with a disproportionately low number of autopsy studies. Diffuse alveolar damage (DAD), the histomorphological correlate of ARDS, has been most frequently reported, as have been thromboembolic events [3,4,5,6,7,8,9,10,11,12,13].

In this study, we document our findings in consecutive autopsies of patients who died of COVID-19 during the first wave, with focus on pulmonary and cardio-vascular pathology including viral detection in formalin-fixed paraffin embedded (FFPE) tissue. We show the high prevalence of DAD as a reaction pattern in COVID-19, the high number of overlying acute bronchopneumonia, and high-level pulmonary virus detection limited to patients who died ≤2 weeks after onset of symptoms, correlating with exudative phases of DAD.

## 2. Materials and Methods

### 2.1. Patients

Our cohort comprised consecutive postmortem examinations of patients who died with COVID-19 disease, performed at the Institute of Pathology of Lausanne University Hospital (CHUV) from the beginning of the pandemic in March 2020 to 31 May 2020. During this period, 77 patients died with COVID-19 at the Lausanne University Hospital (CHUV), of which 12 (16%) were autopsied. Two additional patients autopsied in our study died at external hospitals. In Switzerland, medical autopsies are performed only on clinician’s request and after consent from the family of the deceased. Two of 14 autopsies had to be excluded from our study due to the patients’ refusal to reuse of biological material and clinical data for research purposes. All patients had positive PCR-tests from nasal swabs for SARS-CoV-2 and a clinical diagnosis of COVID-19. Comprehensive information on the onset of symptoms, laboratory tests, treatment and co-morbidities was extracted from clinical files. The control cohort for multiplex imaging comprised patients autopsied before the beginning of the COVID-19 pandemic (2010-January 2019), with matched histomorphological lung pathology findings. This study has been performed according to the Helsinki declaration and following the rules of the local institutional review committee (Ethical Committee of the Canton de Vaud, CER-VD, Switzerland, protocol number 2020-01257, 4 July 2020).

### 2.2. Postmortem Examination

The mean and median time intervals between death and autopsy were 27 h and 14 h, respectively (range: 6.5–70 h). All postmortem examinations were complete body autopsies (including brain examination in 3 cases). The autopsies were conducted in accordance with in-house guidelines for handling infectious cases, rapidly additionally adapted. In short, autopsies were performed in a dedicated COVID-19 autopsy room with defined entry- and exit-areas, and a minimal number of staff involved. All personnel were equipped with FFP3-masks, face shields, disposable gowns, dedicated shoes with disposable shoes covers and multiple layers of disposable gloves.

The tubular digestive tract was only dissected if the autopsy was performed <24 h after death or in case of specific clinical enquiries. Thoracic organs and trachea were eviscerated and separated after external examination. Organ weight was assessed fresh. Lungs were inflated with buffered 4% formalin prior to immersion in large volumes of formalin at room temperature together with other organs. All organs were fixed in buffered formalin for at least 72 h before further dissection. The lungs were sliced into 1 cm thick sagittal sections. The heart was cut into transverse slices after in situ dissection of the coronary arteries. We designed an ad hoc protocol for extensive histological sampling, including standard sampling of 7 blocks from different areas of the heart and coronary arteries, 2 blocks from the trachea (distal and proximal), 14 blocks from the right lung and 10 blocks from the left lung. 

Four to five μm thick sections of routinely processed FFPE tissue blocks were stained with standard hematoxylin and eosin (H&E) and additional stains according to standard protocols. In each case, selected lung samples were stained with Elastica van Gieson, trichrome, periodic acid Schiff (PAS) and Grocott’s methenamine silver stain. Representative heart sections were stained with PAS, trichrome and Congo Red. Lung tissue slides (mean blocks/case: 23, range: 14–43) and heart tissue slides (mean blocks/case: 7, range: 4–16) were reviewed by at least three pathologists (SB, LDL, DM).

### 2.3. Quantitative RT-qPCR for SARS-CoV-2 Detection in FFPE Tissues

Ten μm thick sections from lung and heart FFPE tissue blocks were processed for automated total RNA extraction on the QIAcube instrument (Qiagen, Hilden, Germany) using RNeasy FFPE kit (Qiagen) following manufacturer’s instructions with slight modification (incubation with proteinase K at 56 °C overnight). RNA was eluted with 30 μL of RNase-free water and quantified using Qubit High sensitivity RNA kit (ThermoFisher Scientific, Waltham, MA, USA). 

For SARS-CoV-2 quantification, retro-transcription and real-time PCR (RT-qPCR) were performed on a Cobas z 480 instrument (Roche Diagnostics, Basel, Swizerland), starting from 50 ng (or 250 ng where specified) of total RNA in 20 μL reaction volume, using one-step RT-qPCR LightCycler^®^ Multiplex RNA Virus Master Mix (Roche Diagnostics) following manufacturer’s instructions. The primers used were LightMix^®^ Modular SARS and Wuhan CoV E-gene, LightMix^®^ Modular Wuhan CoV RdRp-gene and LightMix^®^ Modular *MSTN* extraction control (internal control for RNA and RT-qPCR quality) (all from Roche Diagnostics). Quantification cycle (Cq) values for E gene, RdRp and *MSTN* genes were determined with LightCycler^®^ 480 Software release 1.5.1.62 by selecting the second derivative maximum method [14]. For E gene and RdRp copy number evaluation, a standard curve was included in each run using a pool of RNAs from pre-pandemic SARS-CoV-2-negative autopsy samples spiked with known concentrations of E gene or RdRp positive controls (provided with the primer kits, Roche Diagnostics). Each sample was analysed in duplicate and a no template control was included in each run. The limit of detection (LoD) for E gene and RdRp assays was defined as the input copy number with a 95% probability of a positive PCR result [15]. LoD was determined by probit regression analysis with glm function (package stats v.4.1.0 included in R) using a probit model and a binomial distribution to create the model and ggplot2 (v.3.3.3) to generate the plots by analyzing each of 28 replicates of the respective positive control at 4-fold serial dilutions corresponding to 1000, 250, 62.5, 15.6, 3.9 and 0 copies per reaction.

### 2.4. Immunohistochemical Staining

Immunohistochemical staining (IHC) was performed using the Ventana Discovery Ultra Autostainer (Roche Ventana, Tucson, Arizona, USA) following the manufacturer’s instructions. Four μm FFPE sections were initially heated up for 4 minutes at 72 °C and placed in EZ prep solution (#950-102, Roche Ventana) for deparaffinization. Antigen retrieval was performed at 95 °C in Cell Conditioning Solution 1 (CC1, #950-124, Roche Ventana) for 56 minutes. A polyclonal anti-SARS-CoV spike protein antibody (#40150-T62-COV2; Sino Biological, Beijing, China) and a mouse monoclonal anti-SARS-CoV 2 nucleocapsid protein (NP) antibody (#40143-MM05, Sino Biological) were applied at a dilution of 1/250 and 1/100, respectively, for 56 minutes at 36 °C for cross detection of the respective SARS-CoV-2 proteins, followed by incubation with the DISCOVERY OmniMap anti-Rabbit HRP kit (#760-4310, Roche Ventana) or with the DISCOVERY OmniMap anti-Mouse HRP kit (#760-4310, Roche Ventana) for 16 minutes. The DISCOVERY ChromoMap DAB detection kit (#760-159, Roche Ventana) was used as detection system. Tissue counterstaining was performed with Hematoxylin II solution (#790-2208, Roche Ventana). Evaluation was performed independently by two pathologists (SB and LDL) as positive or negative.

### 2.5. In Situ Detection of SARS-CoV-2 mRNA in FFPE Tissues

FFPE tissue blocks were cut into 4 μm sections. Actively transcribed SARS-CoV-2 was detected by RNAScope technology (ACDBio, Newark, CA, USA) using a specific probe for the SARS-CoV-2 S protein (2.5VS Probe-V-nCoV2019-S, Ref 848569, ACDBio). 2.5VS positive control probes Hs-UBC (housekeeping gene positive probe, Ref 312029, ACDBio) as well as 2.5VS negative control probes dapB (nonspecific bacterial gene probe, Ref 312039, ACDBio) were run in every assay to check the tissue and technical quality after processing (Ventana Discovery Ultra Autostainer, Roche Ventana). RNAscope probes were incubated for 4 min. The RNAscope VS Universal HRP Reagent kit (Ref 323200, ACDBio) was used containing all required specific materials for standard target retrieval (CC1 24 min, recommended for lung tissue) and amplification of the targeted mRNA. All the buffers required, Discovery Wash (10×), Ultra LS, SSC Buffer (10×), Reaction Buffer (10×); Discovery CC1, as well as the probe and pre-treatment dispensers were purchased from Roche Diagnostics. For the detection of fluorescence, Discovery Red 610 kit (Ref. 760-245, Roche Diagnostics) was incubated at RT for 32 min. The tissue sections were then counterstained using Discovery DQ DAPI (Ref 760-4196, Roche Diagnostics) at room temperature for 12 min, rinsed in water with soap and mounted using Dako Fluorescent Mounting Medium (Ref S3023, Dako, Santa Clara, CA, USA). Immunofluorescence images were acquired using the Vectra Polaris imaging system (Akoya, Marlborough, MA, USA). All images were recorded using 20x magnification and the Phenochart 1.0.12 software (Akoya) was used for analysis of areas of interest identified on the corresponding H&E section.

### 2.6. Immunofluorescence Staining

Fluorescent multiplex immunostaining was performed to assess inflammatory infiltrates and cell components in the lung of the five selected cases and controls. Briefly, the IHC protocol on the Ventana Discovery Ultra Autostainer (Roche Diagnostics) was used for sequential immunofluorescence (IF) multiplex staining, using antigen retrieval and antibody blocking steps, staining with primary antibodies (Table 1 and Table 2) and thereafter incubation with secondary HRP-labeled antibodies, followed by detection with optimized fluorescent Opal tyramide signal amplification (TSA) dyes (Opal 7-color Automation IHC kit, from Akoya, Ref. NEL821001KT) and repeated antibody denaturation cycles. Data acquisition and analysis was performed as previously described [16].

## 3. Results

### 3.1. Patient Cohort

The patient cohort was comprised of 5 women and 7 men (median age: 73 years; range 35–96) who died 6–38 days after onset of symptoms (median: 14.5 days). For this study, the 12 patients included were ranked according to the duration of symptoms (case # in Table 3). With the exception of one tetraplegic 35-year-old patient who died of acute bacterial aspiration pneumonia and did not show DAD in the lung (case #7), all patients were ≥60 years of age. Eight patients received mechanical ventilation for 6–19 days. All but two patients (cases #8 and #11) died at the Lausanne University Hospital.

Comorbidities comprised most frequently cardiovascular diseases including hypertension (*n* = 9/12), diabetes mellitus (*n* = 4/12), chronic obstructive pulmonary disease (COPD, *n* = 4/12) and chronic renal failure (*n* = 2/12). Four patients had an underlying malignancy (follicular lymphoma, prostate cancer and lung cancer), three of which were autopsy findings (prostate and lung cancer). Secondary pulmonary infections were documented radiologically in 5/12 patients (cases #4, 5, 6, 7 and 8), and suspected clinically in 1 patient (case #3). Six patients received antiviral therapy, 8/12 were anticoagulated, and none of the patients received corticosteroids. Detailed clinical data including co-morbidities and medication are provided as Appendix A.

The control cohort for multiplex imaging—performed on lung tissue of 5 patients from the COVID-19 cohort—comprised lung tissue from 5 patients autopsied before the beginning of the COVID-19 pandemic, 2 women and 3 men (median age: 58 years; range 31–71). Two patients died from ARDS due to infectious exacerbations of chronic interstitial lung disease (idiopathic pulmonary fibrosis), histologically with DAD in the proliferative phase. One patient died of pneumonia due to H1N1 influenza, with DAD in the proliferative phase. Two additional patients showed DAD in the exudative phase, one with beginning organization, without specific infectious agents identified intra vitam nor at autopsy.

### 3.2. Histopathological Findings

#### 3.2.1. Lungs

The lung weights were significantly elevated, with a mean and median combined weight of 1815 g and 1865 g, respectively (range: 960–2560 g; normal: 700–1000 g). Eleven patients showed DAD in different stages of evolution (Table 3, Appendix A, Figure 1). One was in early exudative phase, with congestion, edema and hyaline membrane formation. Six (50%) were in late exudative phase, with additional thickening of alveolar septa and interstitial lympho-plasmacytic infiltrates. Four (33%) were characterized as proliferative phase/organizing DAD, with intraalveolar fibrosis in continuation with the septal interstitium, in one patient presenting as acute fibrinous and organizing pneumonia. The extent of DAD was very patchy and localized, especially in the earlier stages of DAD. As “diffuse” in DAD relates to the diffuse involvement of the alveolar septum, and not to the involvement of the complete lung parenchyma of the lungs, the patchy involvement was no hindrance to the diagnosis of DAD. Acute bronchopneumonia was present in six patients (50%), in one case (case #7) without underlying DAD. One patient (case #11) showed invasive aspergillosis with necrotic bronchitis, meeting the diagnostic criteria for COVID-19-associated pulmonary aspergillosis (CAPA) [17]. Microthrombi were present only in exudative DAD in five patients. One patient (case #8) showed peripheral emboli with associated focal hemorrhagic lung infarction. True necrotizing vasculitis was absent. Focal interstitial and vascular amyloid deposition compatible with senile amyloidosis was present in one patient (case #4), although without detectable cardiac involvement.

#### 3.2.2. Heart

Heart findings were associated with preexisting heart disease. The mean and median heart weights were 466 g and 420 g, respectively (range: 330–870 g; normal: 300–360 g). Myocyte hypertrophy was present in 10/12 patients (83%), and 4/12 (33%) patients showed coronary artery disease with ≥50% stenosis. Patchy ischemic interstitial fibrosis was present in 5/12 cases (42%) and 3/12 patients (25%) had localized scarring, one with heart wall aneurism. Pacemakers were in place in 3/12 (25%) patients and 1 patient had a mechanical aortic valve.

Only one patient (case #12) showed focal perivascular amyloid depositions, notably without associated pulmonary involvement. Basophilic degeneration in a minority of cardiomyocytes (<10%) was present in all patients. There was no myocarditis or vasculitis detected despite extensive sampling. All findings are listed in Table 3 and Appendix A.

#### 3.2.3. Other Significant Findings

Autopsy revealed malignant neoplastic disease in three patients. One patient (case #10) showed a locally advanced pulmonary squamous cell carcinoma with local lymph-node metastases. Adenocarcinoma of the prostate was detected in two other patients (cases #8 and #11), in one of them (case #8) with a concomitant pancreatic neuroendocrine tumor. One patient (case #1) had a history of follicular lymphoma diagnosed 2 years ago, in clinical remission and without residual or recurrent disease in the postmortem examination. All findings are listed in Appendix A.

Moderate to severe systemic arteriosclerosis was present in 10/12 patients (83%).

### 3.3. Detection of SARS-CoV-2 Using RT-qPCR

#### 3.3.1. Establishing the Methodology 

At the time the autopsies were performed, there was no validated assay available for measuring SARS-CoV-2 viral load in FFPE samples by RT-qPCR. We used commercial assays from Roche Diagnostics targeting E gene and RdRp, initially developed for evaluating the expression level of these viral genes in fresh samples (e.g., nasal swabs) [18], and coupled them with a human *MSTN* assay, enabling the evaluation of RNA quality (internal quality control). The E gene assay detects SARS-CoV-2 and SARS-CoV (but not other common human respiratory viruses like MERS-CoV) while the RdRp assay is specific for SARS-CoV-2 [18,19]. Adequate internal controls allowing accurate data assessment are especially important when using FFPE samples, which often yield low quality and/or low quantity RNA. We chose to use human *MSTN* expression level as internal controls to evaluate RNA integrity, retro-transcription efficiency or PCR inhibition, since it is ubiquitously expressed in tissues but at relatively low levels. This latter is important to avoid overestimation of process and sample quality that could arise when using genes that are highly expressed. Since the *MSTN* expression level differs slightly between tissue types (own observations and previously reported), we defined an *MSTN* Cq cut-off value for each tissue type analyzed and each amount of RNA input used, above which the extracted RNA would be considered degraded and the assay inconclusive (if negative for the viral genes). These “quality thresholds”, empirically defined at the 75th percentile (Q3) of all *MSTN* Cq values obtained from SARS-CoV-2-positive samples, were as follows: Cq = 33 for lung (*n* = 9) and Cq = 35 for heart samples (*n* = 10) for assays using 50 ng of RNA; Cq = 32 for lung (*n* = 5) and Cq = 34 for heart samples (*n* = 8) when using 250 ng of RNA. To determine the LoD of E gene and RdRp assays in FFPE samples under our conditions, probit regression analyses were performed on several replicates of appropriately diluted positive controls. The computations defined LoD, with 95% detection probability, at 5.4 copies per reaction for E gene (95% confidence interval (CI): 4.4–7.4) and at 20.8 copies per reaction for RdRp (95% CI: 16.6–29.1) (Appendix A). According to our finally established methodology, we analyzed each sample starting at 50 ng RNA. In case of values < LoD for both viral genes, RT-qPCR was repeated with an RNA input of 250 ng. In case of persistent results < LoD, the *MSTN* Cq value was reevaluated: if it was below the corresponding “quality threshold”, the result was considered negative; if equal or above, it was considered inconclusive. In case of a positive result (≥LoD) for any of the viral genes studied, the viral load was quantified and expressed as copy number per reaction.

#### 3.3.2. Virus Detection Using RT-qPCR in Lung and Heart 

All 12 lung samples and 11/12 heart samples yielded conclusive results. With 50 ng of RNA input, the range of all *MSTN* Cq values was 25.11–34.96 for lung (median 32.72, *n* = 12), and 31.21–38.45 for heart samples (median 34.88, *n* = 12); with 250 ng of RNA input, the range was 23.02–31.84 for lung (median 29.96, *n* = 9) and 28.88–37.10 for heart samples (median 33.71, *n* = 12). 

SARS-CoV-2 was detected in all lung samples (selecting lesional regions for investigation), with very variable viral levels (Table 3). High amounts of virus were present in the 6 patients with exudative DAD and symptom-duration ≤14 days (range E gene: 26,161–537,302 copies per reaction; RdRp: 3129–64,345). The 6 patients with low viral copy levels (range E gene: 11–178 copies per reaction; RdRp: all cases < LoD) were symptomatic for ≥15 days, comprising all 4 cases with organizing DAD, the patient without DAD and one case of exudative DAD.

SARS-CoV-2 was detected in 6/11 contributive heart samples (Table 3). Viral levels were overall much lower than in the matched lungs (range E gene in heart: 0–479 copies per reaction; RdRp: all cases < LoD), without correlation with symptom duration or clinico-pathological features of the underlying heart disease. 

Irrespective of the tissues tested, detection levels of E gene were roughly 10-fold higher than those of RdRp. Accordingly, all samples positive for E gene were also positive for RdRp, but not reciprocally, highlighting the difference in virus detection rate between the 2 assays, as previously reported, and which might be attributed to a mismatch introduced in the reverse primer or to the fact that E gene is present in genomic and subgenomic RNA, wheras RdRp is only present in genomic RNA [19,20,21].

### 3.4. Detection of SARS-CoV-2 Using Immunohistochemistry

Lung tissue sections representative of the main lesional pattern of all 12 patients were stained with both the anti-SARS-CoV spike protein antibody and the nucleocapsid protein antibody. Both assays showed concordant results, with positive staining in all 6 patients with high-viral load as assessed by RT-qPCR, associated with exudative phase of DAD and symptom duration <14 days. Concordant with literature, positivity was granular and pronounced in hyaline membranes and intraalveolar cells, well corresponding to pneumocytes and macrophages [4,11,22,23]. We could not detect unequivocal positivity in respiratory epithelium, which has been described as very focal by others [22]. The lungs of all 6 patients with low viral load (RT-qPCR) were negative. There was complete concordance of the independent assessment of the stainings by two pathologists.

### 3.5. Multiplex Imaging and In Situ Detection of SARS-CoV-2 mRNA

Lung tissue sections of 5 patients were analyzed with a multiplex imaging assay allowing for the simultaneous detection of ACE2, the main receptor for SARS-CoV-2 [24,25], and adaptive and innate immune cell types (CD3 T cells and CD68 macrophages) (Table 1). Most epithelial cells (PANCK+) and a subset of macrophages (CD68+) expressed varying degrees of ACE2 (Figure 2A,B). Given recent reports on lung infiltration by lymphocytes (CD4 T cells) in COVID-19 patients [3,26,27], we sought to investigate the prevalence of tissue CD4 and CD8 T cell populations in our cohort. Quantitative imaging analysis of control tissue (random lungs with DAD from postmortem examinations performed before the emergence of SARS-CoV-2) and the COVID-19 patient cohort showed no difference when total CD3+ lymphocytes (expressed as a frequency of total tissue imaged cells) were analyzed (*p* = 0.87, Figure 2C). We observed a slightly higher CD4 to CD8 T-cell ratio in the COVID-19 patient group compared to the controls, although it lacked statistical significance (*p* = 0.55; Figure 2D).

Next, the presence of actively transcribed virus was investigated by RNAscope in 5 selected cases from the cohort (cases #1, #3, #4, #6 and #10) (Figure 3A) while a sequential tissue section was subjected to multiplex imaging (panel 2, Table 1) for the identification of possible cellular localizations of the virus (Figure 3B). In agreement with literature [4], we observed an extensive SARS-CoV-2 mRNA positive signal associated with hyaline membranes (Figure 3C). The prevalence of RNAscope positive events was in line with the viral titers measured by RT-qPCR. Analysis of the sequential multiplex imaged sections confirmed the expression of ACE2 receptor in cells positive for SARS-CoV-2 mRNA signal and we identified pneumocytes (PANCK+) and macrophages (CD68+) harboring viral transcripts (Figure 3B).

## 4. Discussion

We report autopsy findings of 12 consecutive COVID-19 patients, focusing on lung and cardiovascular pathology, including virus detection using different methodological approaches. Most of our findings are in line with previously published observations, but some aspects merit in depth discussion. 

The clinical and epidemiological characteristics of our cohort were in accordance with literature [4,8], showing a slight male predominance of 58%, a median age of 73 years, with only one patient younger than 69 years-of-age, and multiple comorbidities. 

We confirm the high prevalence of DAD in patients with fatal COVID-19 in our patient cohort. DAD is the morphological pattern of acute lung injury due to any cause, and one of the classic patterns of interstitial lung diseases [28,29]. As expected from its temporally preserved progression from the exudative phase, via the organizing/proliferative phase to a fibrotic phase [30], early/exudative stages of DAD were associated with a shorter disease duration, and proliferative DADs presented in patients with longer standing disease, with a cut-off at around 2 weeks since onset of symptoms. 

Early DAD stages were associated with a high viral load detected using RT-qPCR, presence of SARS-CoV spike protein and nucleocapsid protein detected using immunohistochemistry and actively transcribed SARS-CoV-2 confirmed by in-situ detection of SARS-CoV-2 mRNA (RNAscope assay). In patients who died 2 weeks or later after onset of symptoms and presented organizing stages of DAD, only the highly sensitive RT-qPCR assay could confirm the presence of low copy numbers of virus in the lungs. This mirrors previous reports of smaller patient cohorts [11,22,31], although other groups reported virus detected in individual patients who died after 4 weeks of ongoing disease, using immunohistochemistry confirmed by RNA-ISH [4]. Those discrepancies could be explained by the heterogeneity of lung affection and a possible re-infection of adjacent areas leading to longer virus detection.

Our in situ analysis showed a wide expression of ACE2, the major SARS-CoV-2 receptor, in epithelial cells and interstitial macrophages, a profile that could further support local viral replication. We detected macrophages positive for SARS-CoV-2 mRNA, in line with recently published data [22,32], raising the possibility that these cells could affect the course of the disease at two levels: by contributing to local viral spread and by producing pro-inflammatory mediators [33]. The lung tissue of our COVID-19 cohort harbored a similar frequency of CD3 T cells as the non-COVID-19-DAD control group, with the majority of T cells expressing CD4 in both cohorts. However, we observed slightly higher ratios of CD4 to CD8 in COVID-19 samples. Although this could indicate a preferential recruitment of CD4 T cells in the inflamed tissues tested, the result has to be interpreted with caution due to the lack of statistical significance, the small sample size and the differing patient characteristics between the two groups tested, e.g., secondary acute bronchopneumonia in 2/5 and underlying neoplasms in 2/5 COVID-19 patients. Although we could not observe a preferential localization of CD3 T cells in proximity to macrophages, it is likely that monocytes/macrophages play a role in recruiting T cells into the lungs [32].

The pattern of DAD reflects acute lung injury, and histomorphologic particularities reliably predicting COVID-19 as an underlying etiology have not yet been established [6,31]. The “diffuse” in DAD alludes to all parts of the alveolar septum being damaged, including endothelial and alveolar lining cell injury. Microthrombi are characteristic features of DAD since its earliest descriptions [34]. Although endotheliitis [35], microthrombi, intusscusceptive angiogenesis [36] and peculiar compositions of associated inflammatory infiltrates have been described in deceased patients with COVID-19 with the claim of “uniqueness”, those were very small case series and the finding are principally compatible with unspecific DAD. In a small series comparing COVID-19 induced DAD and matched non-COVID-19 controls, a group of 3 dedicated pulmonary pathologists was unable to detect morphological differences [6]. Concordantly, another group of specialized pathologists showed nonspecific clinical and pathologic changes shared between severe cases of COVID-19 and seasonal and pandemic influenza [7]. 

We validated the high frequency of associated or superimposed acute bronchopneumonia previously described in deceased COVID-19 patients [4,9], which was present in over half of the patients in our cohort. Ulcerative trancheobronchitis independent from bronchopneumonia has also been claimed as a prominent finding [4]. Only the one patient presenting with COVID-19-associated pulmonary aspergillosis (CAPA) showed a localized ulcerative tracheobronchitis in our cohort. Invasive mold disease has been described in up to 20% of COVID-19 autopsy cohorts [37]. Contrary, it was recently shown to be present in 2% of patients in a systematic review of autopsies [38], a discrepancy which might be due to the extent of tissue sampling.

Many groups have described frequent thromboembolic events as autopsy findings in COVID-19 patients [4,5,9,37,39]. In our cohort, it was only one patient who presented with multiple peripheral emboli with an associated hemorrhagic infarction. He had two neoplastic diseases (adenocarcinoma of the prostate and a small neuroendocrine tumor of the pancreas, <1 cm), which are known to result in a hypercoagulative state [40]. Microthrombi were present in 3/12 patients.

Heart findings consisted of hypertrophy, ischemia and coronary arterial disease, associated with cardiac co-morbidities, but showed no SARS-CoV-2 associated lymphocytic inflammation. Only 1/12 cases showed senile amyloidosis, despite extensive sampling and application of special staining for amyloid to each case. Menter et al. report senile amyloidosis as confirmed by immunohistochemistry for ATTR in 6/21 (29%) of their autopsied COVID-19 patients, all aged between 76–96 years, with additional amyloid deposition in pulmonary vessels in three of them (overall: 14%) [39]. Although this number was higher than in an age-matched cohort from the files of the same institution, and amyloidosis was therefore claimed to be linked to a fatal COVID-19 outcome, the higher proportion could also be explained by a more careful evaluation in COVID-19 cases. In previous autopsy studies the prevalence of senile amyloidosis reached 65% in patients aged >90 years and 11.5–25% in patients older than 80 years [41,42,43]. Extracardiac involvement in senile amyloidosis typically involves pulmonary arteries and alveolar septa, in line with the findings by Menter et al. Only one patient in our cohort showed focal pulmonary interstitial and vascular amyloid deposition compatible with senile amyloidosis, although without detectable cardiac involvement. We detected no myocyte necrosis, thrombosis of small vessels, acute myocardial infarction or lymphocytic or eosinophilic myocarditis, which were described in 1–7% of 99 COVID-19 autopsies as recently summarized [8].

## 5. Conclusions

In summary, in our autopsy cohort of 12 COVID-19 patients we confirm the high prevalence of DAD as a reaction pattern in fatal COVID-19, the high number of overlying acute bronchopneumonia and high-level pulmonary virus detection limited to patients who died ≤2 weeks after onset of symptoms, correlating with exudative phase of DAD. In those patients, we detected actively transcribed SARS-CoV-2 in pneumocytes and macrophages, without increased T-lymphocyte infiltrations as compared with non-COVID-19 DAD controls. Only very sensitive RT-qPCR morphology could detect low virus levels in patients with long-standing disease, correlating with organizing phases of DAD. 

Our frequencies of thromboembolic events, cardiac amyloidosis or lymphocytic infiltration of heart tissue were lower than reported. 

The limitation of our study is the small sample size and a possible selection bias, as autopsy was allowed and performed in only 16% of patients who died during the time period in our University Hospital. It is only after larger comparison studies using adequate control cohorts have been conducted that we will be able to answer the question of specificity of the hitherto published morphological aspects of COVID-19.

## Figures and Tables

**Figure 1 diagnostics-11-01357-f001:**
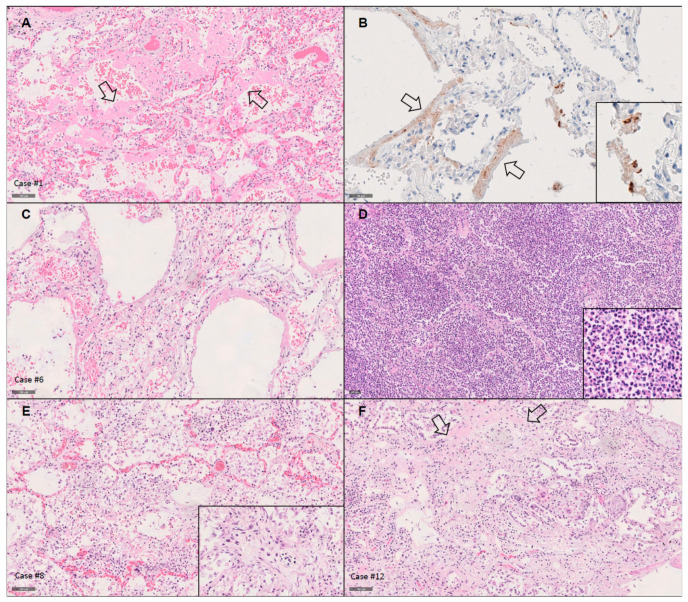
Lungs showed diffuse alveolar damage in different stages of evolution. (**A**) Acute exudative phase with extensive hyaline membranes formation (arrows), without thickening of alveolar septa (case #1; H&E, ×100). (**B**) In the patients with exudative phase diffuse alveolar damage, SARS-CoV-2 antigen was present in the hyaline membranes (arrows) and scattered cells (inset), as exemplified in case #1 using the nucleocapsid protein antibody (×200). (**C**) In the late exudative phase some septal thickening and focal early organization was present apart from extensive hyaline membrane formation (case #6; H&E, ×100). (**D**) Half of our cohort had superimposed (bacterial) acute bronchopneumonia, as exemplified in case #6 (H&E, ×100; inset with higher magnification, highlighting densely packed granulocytes). (**E**) Proliferative phase presented with fibrosis, obliterating alveolar lumina, as highlighted in the inset (case #8; H&E, ×100). (**F**) The density and collagen richness of fibrosis (arrows) increased with longer-standing disease, as exemplified in case #12 (H&E, ×100).

**Figure 2 diagnostics-11-01357-f002:**
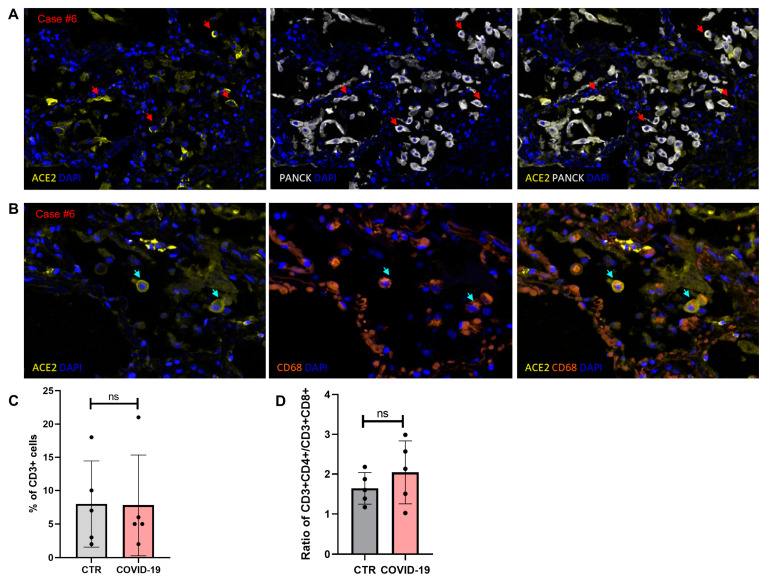
(**A**,**B**) Multiplex imaging analysis depicting expression of ACE2, the major receptor for SARS-CoV-2, in different cellular compartments of lung tissues in a representative patient with COVID-19 (case #6). (**A**) Red arrows point to ACE2-expressing pneumocytes, defined by double-expression of ACE2 (yellow) and PANCK (white). (**B**) Cyan arrows highlight ACE2-expressing macrophages, defined by double-expression of ACE2 (yellow) and CD68 (red). All images were acquired on the Vectra Polaris microscope using a 20x objective magnification, and zoomed in areas are depicted for visualization of macrophages. (**C**) The was no difference in the relative frequencies (% of total imaged cells) of CD3+ T cells in lung control tissues (non-COVID-19 DAD, *n* = 5) and COVID-19 lung tissues (*n* = 5), expressed as mean values with SD at the scatter dot plots (Mann–Whitney U test, *p* = 0.87). (**D**) There was no statistically significant difference of the CD4+ to CD8+ T lymphocyte ratio between the two cohorts (Mann–Whitney U test, *p* = 0.55).

**Figure 3 diagnostics-11-01357-f003:**
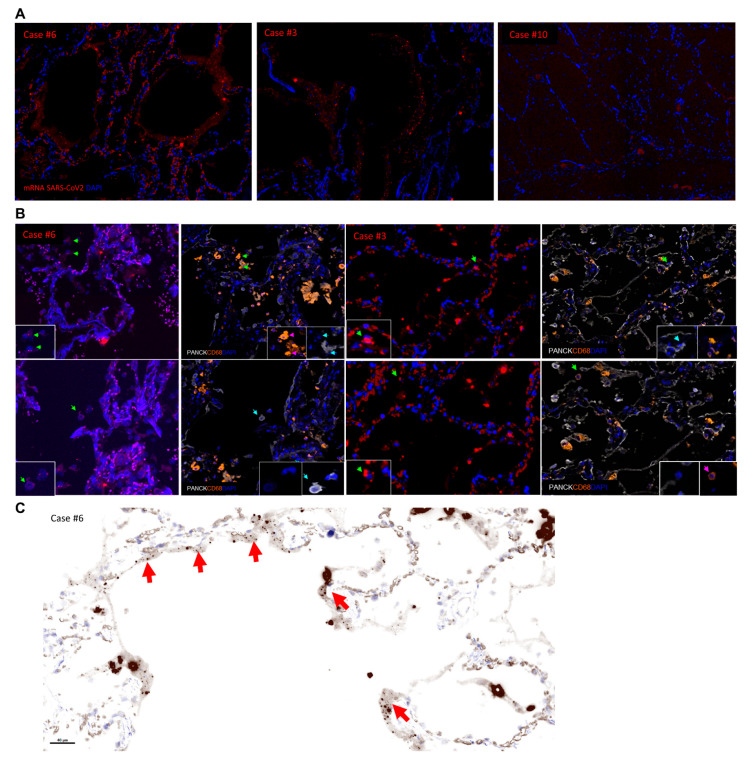
Detection of actively transcribed SARS-CoV-2 in different cellular compartments of lung tissue. (**A**) In situ expression of SARS-CoV-2 mRNA (red), detected by RNAscope, in lung tissues from three infected patients with different levels of SARS-CoV-2 viral loads as detected by RT-qPCR in FFPE lung tissue. Granular positivity highlights hyaline membranes in RT-qPCR highly positive cases #3 and #6. Lack of in situ expression of SARS-CoV-2 mRNA in case #10, which showed also very low SARS-CoV-2 copy numbers in the RT-qPCR assay. (**B**) SARS-CoV-2 mRNA expression in pneumocytes and macrophages in two representative patients (cases #3 and #6), as detected by RNAscope and multiplex imaging on sequential lung tissue sections (magnifications as insets). Green arrows point to cells positive for viral mRNA. Cyan arrows highlight SARS-CoV-2 mRNA positive pneumocytes, defined by PANCK-expression. Magenta arrows depict SARS-CoV-2 mRNA positive macrophages, defined by CD68-positivity. (**C**) Lung tissue (case #6) stained for SARS-CoV2 mRNA expression (RNAscope). A simulated H&E image (generated from a fluorescent image processed using the inForm software) shows tissue areas positive for SARS-CoV-2 mRNA signal (red arrows) mainly associated with hyaline membranes. All images were acquired on the Vectra Polaris microscope using a 20× objective magnification.

**Table 1 diagnostics-11-01357-t001:** Antibodies used for immunofluorescence staining.

Antibody	Clone	Tissue Specificity	Species	Source	Dilution
ACE2	CL4035	Membranous	Mouse	Atlas Antibodies/ AMAB91262	1:1000
CD3	2GV6	Cytoplasmic/ Membranous	Rabbit	Roche/dispenser	Prediluted
CD4	SP35	Membranous	Rabbit	Roche/dispenser	Prediluted
CD8	C8/144B	Membranous	Mouse	DAKO/M7103	1:30
CD68	PG-M1	Cytoplasmic	Mouse	DAKO/M0876	1:200
PANCK	AE1/AE3	Cytoplasmic	Mouse	DAKO/M3515	1:100

ACE2, angiotensin converting enzyme 2 receptor; CD3, cluster of differentiation 3; CD4, cluster of differentiation 4; CD8, cluster of differentiation 8; CD68, cluster of differentiation 68; PANCK, pan cytokeratin.

**Table 2 diagnostics-11-01357-t002:** Composition of antibodies used in the multiplexed immunofluorescence panels.

Panel	Antibodies	Opal Fluorophores	Dilution
Multiplexed IF Panel 1	CD3	Opal 520	1/400
	CD4	Opal 690	1/150
	CD8	Opal 620	1/150
	CD68	Opal 480	1/700
	PANCK	Opal 780	1/25
	DAPI	Spectral DAPI	
Multiplexed IF Panel 2	ACE2	Opal 620	1/150
	CD3	Opal 520	1/400
	CD68	Opal 480	1/700
	PANCK	Opal 780	1/25
	DAPI	Spectral DAPI	

ACE2, angiotensin converting enzyme 2 receptor; CD3, cluster of differentiation 3; CD4, cluster of differentiation 4; CD8, cluster of differentiation 8; CD68, cluster of differentiation 68; PANCK, pan cytokeratin.

**Table 3 diagnostics-11-01357-t003:** Patient cohort including pulmonary, cardio-vascular findings and virus detection in lung and heart tissue obtained during postmortem examination. Patients are ranked according to duration of symptoms.

Case	Gender, Age (Years)	Duration of Symptoms {Mechanical Ventilation} (Days)	Time Interval between Death and Autopsy (h)	Lung Findings *		Virus Detection (Lung)				Cardiac Findings	Virus Detection (Heart)	
				Diffuse Alveolar Damage (DAD)	Acute Broncho-pneumonia	RT-qPCR (Viral Copy Number per Reaction)		Viral RNA (RNA Scope)	IHC ^§^		RT-qPCR (Viral Copy Number per Reaction)	
						RdRp LoD: 20.8c/r	E Gene LoD: 5.4c/r				RdRp LoD: 20.8c/r	E Gene LoD: 5.4c/r
1	F, 72	6 {-}	14.5	exudative phase	-	64,345	482,526	present	positive	heart 480 g; hypertrophy, patchy fibrosis	<LoD ‡	<LoD ‡
2	F, 72	8 {-}	13	exudative phase	upper lobe predominant, focally with aspirate	3129	26,161	ND	positive	heart 420 g; biventricular hypertrophy; CAD with stenosis up to 70%; fibrotic scar left ventricle (0.7 cm)	<LoD ‡	48 ‡
3	M, 96	8 {-}	6.5	late exudative phase	present in all lobes, necrotizing	18,858	316,089	present	positive	heart 400 g; hypertrophy; CAD with stenosis up to 30%; patchy fibrosis	<LoD	479
4	M, 86	10 {3}	70	late exudative/early proliferative phase	-	76,722	537,302	present	positive	heart 420 g; hypertrophy; CAD with stenosis up to 50%; low-grade diffuse interstitial fibrosis	<LoD	50
5	F, 74	11 {5}	70	exudative phase	present in all lobes	3669	26,874	ND	positive	heart 330 g; hypertrophy; pacemaker in place	<LoD ‡	<LoD ‡
6	F, 71	14 {-}	9.5	late exudative phase and focally proliferative phase	present in all lobes	5469	230,265	present (high)	positive	heart 410 g; status post myocardial infarction with a 1.5 cm scar (apical posterior left ventricle)	<LoD	16
7	M, 35	15 {2}	20	-	present in all lobes, necrotizing (clinical/radio-logical picture: aspiration pneumonia)	<LoD ‡	178 ‡	ND	negative	heart 360 g	<LoD ‡	135 ‡
8	M, 79	16 {4}	61.5	proliferative phase	-	<LoD	69	ND	negative	heart 390 g; hypertrophy; focal fibrosis in one left papillary muscle	<LoD ‡	<LoD ‡
9	M, 75	17 {5}	20.5	late exudative phase	-	<LoD	26	ND	negative	heart 870 g; hypertrophy; signs of chronic ischemia with patchy fibrosis; mechanical aortic valve; pacemaker in place	<LoD ‡	111 ‡
10	F, 73	21 {14}	12	AFOP	- (microbiology: *proteus mirabilis*)	<LoD ‡	11‡	absent	negative	heart 510 g; hypertrophy; CAD with stenosis up to 30%	NC	NC
11	M, 60	18 # {17}	13	proliferative phase	focal, with bronchial ulceration (aspergillosis)	<LoD ‡	24 ‡	ND	negative	heart 470 g; hypertrophy; CAD with stenosis up to 80%	<LoD ‡	<LoD ‡
12	M, 69	38 # {19}	13.5	proliferative phase	focal (microbiology: *serratia marcescens*)	<LoD ‡	49 ‡	ND	negative	heart 540 g; hypertrophy; focal amyloidosis; CAD with stenosis up to 50% and two scar regions (2 cm each) and aneurysm; pacemaker in place	<LoD ‡	<LoD ‡

# Duration of symptoms unknown, the number of days provided is the duration of hospital treatment. * significant lung findings additionally included a pulmonary squamous cell carcinoma (aT3; aN1, aM0) in case #10. § both the anti-SARS-CoV spike protein antibody and the nucleocapsid protein antibody showed concordant results. ‡ Samples for which 250 ng of total RNA were used for SARS-CoV-2 detection by RT-qPCR. AFOP, acute fibrinous and organizing pneumonia; LoD, limit of detection as determined by probit analysis to be 5.4 copies per reaction (c/r) for E gene assay and 20.8 c/r for RdRp assay; ND, not determined; NC, non conclusive.

## Data Availability

The data presented in this study are available on request from the corresponding author. The slides are not publicly available due to technical and privacy reasons.

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
