# Peer review of "Postmortem Cardiopulmonary Pathology in Patients with COVID-19 Infection: Single-Center Report of 12 Autopsies from Lausanne, Switzerland"

_diagnostics, 2021, doi:10.3390/diagnostics11081357_

Round 1

Reviewer 1 Report

I read with interest the article entitled " Postmortem cardiopulmonary pathology in patients with COVID-19 infection: single-center report of 12 consecutive autopsies from Lausanne, Switzerland.”.

The work describes postmortem findings in twelve patients who died from COVID-19. The paper is of great interest and well written. I propose some small correction.

Major Concerns

  • Line 269 The proposed valuation method is acceptable. The evaluation of a method, however, requires an evaluation of further parameters such as calculation of the sample size based on the error, inter and intra-individual repeatability, comparison with different centers. I request a small terminological correction from the authors.
  • Among the conclusive proposals as well as an enlargement of the sample and the establishment of a control group the authors should describe the nature of the control group. A work of great value and scientific value could for example include autopsy cases positive for SARS-Cov-2 but asymptomatic [i.e. Nioi, Matteo, et al. "Autopsies and Asymptomatic Patients During the COVID-19 Pandemic: Balancing Risk and Reward." Frontiers in Public Health 8 (2020).]. This data should be made explicit.
  • Authors should update the bibliography with similar recently published works.

Minor Concerns

  • Line 66 You should clarify whether 12 or 14 autopsies have been done on COVID-19 patients. The text shows that 14 autopsies were performed but two patients refused to provide the data. This should emerge both in this line [12 (16%)] and in the title (12 consecutive…).

Overall I liked the work very much and I ask the authors to meet the review requests as soon as possible to speed up the Editorial evaluation.

Author Response

Response to Reviewer 1 Comments

I read with interest the article entitled " Postmortem cardiopulmonary pathology in patients with COVID-19 infection: single-center report of 12 consecutive autopsies from Lausanne, Switzerland.”.

The work describes postmortem findings in twelve patients who died from COVID-19. The paper is of great interest and well written. I propose some small correction.

Major Concerns

Point 1: Line 269 The proposed valuation method is acceptable. The evaluation of a method, however, requires an evaluation of further parameters such as calculation of the sample size based on the error, inter and intra-individual repeatability, comparison with different centers. I request a small terminological correction from the authors.

Response 1: We thank the reviewer for notifying us regarding this important point. As suggested by the reviewer and in order to comply with precise usage of language, we have deleted “validating” from the title of section 3.3.1 (highlighted in yellow): “Establishing and validating the methodology” (line 295).

Point 2: Among the conclusive proposals as well as an enlargement of the sample and the establishment of a control group the authors should describe the nature of the control group. A work of great value and scientific value could for example include autopsy cases positive for SARS-Cov-2 but asymptomatic [i.e. Nioi, Matteo, et al. "Autopsies and Asymptomatic Patients During the COVID-19 Pandemic: Balancing Risk and Reward." Frontiers in Public Health 8 (2020).]. This data should be made explicit.

Response 2: We thank the reviewer for the insightful comment. As suggested, we have described the control cohort used for the comparison of the CD4/CD8 ratio in our COVID-19 patient collective in more detail under “2.1 Patients” (lines 73-75) and “3.1 Patient cohort” (lines 208-216).

We are very grateful for the reviewer’s suggestion to study of an additional autopsy cohort of SARS-CoV-2 positive but asymptomatic patients, which we will include into our consideration for future projects, as we do not have it available among the patients autopsied during the first wave of the pandemic reported in the present manuscript.

Point 3: Authors should update the bibliography with similar recently published works.

Response 3: We thank the reviewer for carefully studying our work. Pertinent studies are published at a fast pace. As suggested and due, we have updated the bibliography by recently published works.

References added:

Cell Res. 2021 Jun 16;1-11. doi: 10.1038/s41422-021-00523-8. Online ahead of print. A cohort autopsy study defines COVID-19 systemic pathogenesis. Xiao-Hong Yao et al. PMID: 34135479

Int J Mol Sci. 2021 Jun 18;22(12):6558. doi: 10.3390/ijms22126558. Studying SARS-CoV-2 with Fluorescence Microscopy. Lidia V Putlyaeva et al. PMID: 34207305

Lancet Microbe. 2021 Jun 23. doi: 10.1016/S2666-5247(21)00091-4. Online ahead of print. Invasive mould disease in fatal COVID-19: a systematic review of autopsies. Brittany E Kula et al. PMID: 34189490

Microb Biotechnol. 2021 May 16;10.1111/1751-7915.13828. doi: 10.1111/1751-7915.13828. Online ahead of print. SARS-CoV-2 RNA screening in routine pathology specimens Saskia von Stillfried et al. PMID: 33993637

Mycoses. 2021 Jun 22. doi: 10.1111/myc.13342. Online ahead of print. Proven COVID-19-associated pulmonary aspergillosis in patients with severe respiratory failure. Francesco Fortarezza et al. PMID: 34157166

Pathologe. 2021 Mar 26;1-8. doi: 10.1007/s00292-021-00920-1. Online ahead of print. Detection methods for SARS-CoV-2 in tissue Saskia von Stillfried et al. PMID: 33770236

Sci Rep. 2020 Dec 14;10(1):21894. doi: 10.1038/s41598-020-78949-0. Development of immunohistochemistry and in situ hybridisation for the detection of SARS-CoV and SARS-CoV-2 in formalin-fixed paraffin-embedded specimens. Fabian Z X Lean et al. PMID: 33318594

Minor Concerns

Point 4: Line 66 You should clarify whether 12 or 14 autopsies have been done on COVID-19 patients. The text shows that 14 autopsies were performed but two patients refused to provide the data. This should emerge both in this line [12 (16%)] and in the title (12 consecutive…).

Response 4: We value the reviewer’s careful evaluation of our work and her/his suggestions to increase the exact usage of language and definition. In fact, the autopsy COVID-19 cohort included 12 patients from the university hospital and 2 patients from external hospitals. This summed up to 14 consecutive autopsies, of which data can only be reported from 12 cases (due to refusal from the patients). This was clarified in the material and methods section of the revised manuscript as listed below.

In order to comply with precision, as 2 autopsies of the 14 consecutive once could not be reported in the paper due to refusal of 2 patients, we have deleted “consecutive” from the title, as suggested by the reviewer.

Adaptation of the description in line 66 (highlighted in yellow):

“During this period, 77 patients died with COVID-19 at the Lausanne University Hospital (CHUV), of which 12 (16%) were autopsied. Two additional patients autopsied in our study died at external hospitals.”

Reviewer 2 Report

This study reports postmortem cardiopulmonary pathologic findings of 12 patients diagnosed with Covid19 infection. The authors used various covid19 virus detection methods to confirm the presence of virus in the pulmonary or cardiac tissues and investigated the relevance  among clinical findings, disease course, pathologic findings, and viral load. However, it reads just descriptive and difficult to find out close association or clinical relevance among different findings. Although these postmortem pathologic findings of  Covid19 patients are important and worth to be reported, this report could be more improved to be more informative by classify the data according to the important findings or factors (e.g. patterns of diffuse alveolar damage, bronchoalveolar pneumonia or viral loads in the lung tissues) and supplementing more detailed clinical data (e.g., epidemiology, diagnosis, comorbidity, presence of coinfection, treatment). In addition, one of the major findings is higher viral load in patients showing short duration of symptoms. The patients who had a longer duration of symptom might have had lower viral load due to longer time period between the start day of infection or symptoms and the day of death, which could be significant confounding factor. Also, anti-viral treatment can affect viral load in the tissues, and the authors should consider these major confounding factors before making conclusions.

  1. the patients who had high viral load had short symptom duration (<14 days), and the patients with low viral load had long symptom duration (>14 days).  Is it because of the time difference of diagnosis  (postmortem diagnosis of covid19) between rapid progressed patients and relatively slowly progressed patients?  After 14 days of infection, viral load could be decreased, so the patients who survived more than 14 days might have lower viral copy levels compared with newly diagnosed patients.
  2. please describe more detailed clinical information of the patients- antiviral treatment, comorbidity, and co-infection, covid19 diagnosis, et al.
  3. In table 3, if the individual patients data were classified accordig to the major findings- pathologic findings or clinical findings (rapidly progressed patients vs. slowly progressed patients), it would be more informative.
  4.  what about CD68 macrrophage staining results in patients?

Author Response

Response to Reviewer 2 Comments

This study reports postmortem cardiopulmonary pathologic findings of 12 patients diagnosed with Covid19 infection. The authors used various covid19 virus detection methods to confirm the presence of virus in the pulmonary or cardiac tissues and investigated the relevance among clinical findings, disease course, pathologic findings, and viral load. However, it reads just descriptive and difficult to find out close association or clinical relevance among different findings. Although these postmortem pathologic findings of Covid19 patients are important and worth to be reported, this report could be more improved to be more informative by classify the data according to the important findings or factors (e.g. patterns of diffuse alveolar damage, bronchoalveolar pneumonia or viral loads in the lung tissues) and supplementing more detailed clinical data (e.g., epidemiology, diagnosis, comorbidity, presence of coinfection, treatment). In addition, one of the major findings is higher viral load in patients showing short duration of symptoms. The patients who had a longer duration of symptom might have had lower viral load due to longer time period between the start day of infection or symptoms and the day of death, which could be significant confounding factor. Also, anti-viral treatment can affect viral load in the tissues, and the authors should consider these major confounding factors before making conclusions.

Point 1: the patients who had high viral load had short symptom duration (<14 days), and the patients with low viral load had long symptom duration (>14 days).  Is it because of the time difference of diagnosis (postmortem diagnosis of covid19) between rapid progressed patients and relatively slowly progressed patients?  After 14 days of infection, viral load could be decreased, so the patients who survived more than 14 days might have lower viral copy levels compared with newly diagnosed patients.

Response 1: We thank the reviewer for the insightful comment. We fully agree with the reviewer’s interpretation of our findings. Patients who had a short duration of symptoms until death (<14 days) showed the acute phase (=exudative phase) of DAD in the lung, and high viral tissue load. Patients with a longer duration of symptoms until death (>14 days) showed the chronic phase (=proliferative phase) of DAD in the lung. As DAD is a time-preserved mechanism always starting with the exudative phase, the proliferative morphology mirrors longer disease duration of DAD.

We also confirm with our data the detectability of only minimal amounts of virus in the chronic DAD-phase, pointing towards a clearing of virus from the lung in later stages. This goes well with the widely kept belief that the initial damage induced by the virus is perpetuated as a vicious circle by the immune response of the host, also after the virus itself has been cleared from the tissue. In fact, this is the established mechanism of DAD also due to other causes.

The specific immune response to COVID-19 also in comparison with DAD due to other etiologies will be the focus of subsequent works on the cohort (please refer also to Response 4.)

Point 2: please describe more detailed clinical information of the patients- antiviral treatment, comorbidity, and co-infection, covid19 diagnosis, et al.

Response 2: We thank the reviewer for pointing out the importance of clinical information including antiviral treatment, comorbidities and co-infection. Additional relevant information on comorbidities, specific antiviral treatment and bacterial complications have been added in the main text of the revised manuscript (lines 201-206). Detailed clinical information is provided in the supplementary table S1.

Data on additional heart findings and lung findings at autopsy including organ weight, pleural effusions and co-infection is provided in supplementary table S2.

We do not judge our cohort suitable to draw epidemiological conclusions, as it comprises only cases that have been autopsied and therefore includes only a minority of patients who died at the Lausanne University Hospital in the respective period (12 of 77; 16%).

Point 3: In table 3, if the individual patients data were classified according to the major findings- pathologic findings or clinical findings (rapidly progressed patients vs. slowly progressed patients), it would be more informative.

Response 3: We are very grateful for the critical assessment by the reviewer. We fully agree with the reviewer that structuring the table not chronologically (by date of autopsy) but by e.g. clinical or pathological findings is scientifically preferable and increases readability. It gives an associative order to the descriptive data and visualises nicely the associations gained as results from the data of our descriptive study.

In fact, we have classified the patients according to clinical findings (duration of symptoms), which mirrors pace of progression to the common endpoint death (and autopsy). As DAD is a very time-conserved process, but may be patchy and heterogeneous presumably due to re-infection of adjacent areas of the lung, this classification also mirrors well the histomorphological findings in the lung, and interestingly also the extent of virus detected in the lung tissue.

To clarify that the patients are ordered according to duration of symptoms, we have included this information in the manuscript (lines 192-194) and in the title of table 3 in the revised manuscript (line 264).

Point 4: what about CD68 macrophage staining results in patients?

Response 4: We are very grateful for the reviewer’s suggestion to investigate the immune cell composition in more detail, including the characterisation of macrophages. In fact, this work is currently being done in great detail and will be the content of a separate focused manuscript based on this patient cohort, as it is beyond the scope of the present work that focuses on virus detection associated with cardio-pulmonary histomorphology and clinics.

Round 2

Reviewer 1 Report

Congratulations to the authors I think they have greatly improved the paper.

Author Response

We thank the reviewer for this positive and motivating evaluation of our work.